# Effect of Preoperative Oral Antibiotics for Prevention of Incisional Surgical Site Infection After Colorectal Surgery: A Propensity Score Matching Study

**DOI:** 10.3390/medicina60121970

**Published:** 2024-11-29

**Authors:** Ryo Nakanishi, Heita Ozawa, Naoyuki Toyota, Minori Mise, Shin Fujita

**Affiliations:** Department of Colorectal Surgery, Tochigi Cancer Center, 4-9-13 Yohnan, Utsunomiya 320-0834, Tochigi, Japan; heiozawa@tochigi-cc.jp (H.O.); natoyota@tochigi-cc.jp (N.T.); tsc.ninja400@gmail.com (M.M.); sifujita@tochigi-cc.jp (S.F.)

**Keywords:** surgical site infection, oral antibiotics, colorectal surgery

## Abstract

*Background and Objectives:* Recent findings suggest that combining mechanical bowel preparation (MBP) and preoperative oral antibiotics (OAs) decreases the risk of incisional surgical site infections (iSSIs) in colorectal surgery; however, this finding remains controversial. This study examined the efficacy of OAs and MBP in colorectal surgery using propensity score matching (PSM). *Materials and Methods*: Between January 2015 and December 2020, 559 patients with colorectal tumors underwent MBP followed by colorectal surgery. The multivariate analysis used a COX proportional hazards model to extract risk factors for iSSI. PSM was performed to balance the impact of potential co-factors of OAs with MBP (OA) and MBP alone (non-OA) on superficial SSI incidence. *Results*: The multivariate analysis identified non-OA use as an independent risk factor for iSSIs (odds ratio [OR]: 2.44, 95% confidence interval [CI]: 1.22–4.88, *p* = 0.0112). After matching the cohort, both OA and non-OA groups were divided into 217 cases each. The incisional SSI rate was significantly lower in the OA group (*n* = 338) than in the non-OA group (1.61% vs. 5.07%; OR 3.4; 95% CI; 0.123–0.707; *p* = 0.0062). *Conclusions*: This study revealed that OAs with MBP markedly reduced SSI rates. OAs with MBP should be adopted in colorectal surgery.

## 1. Introduction

Postoperative complications associated with colorectal cancer can lead to increased medical costs and extended hospital stays, which increase the burden on both patients and medical staff [1]. An incisional surgical site infection (iSSI) is the most common complication of gastrointestinal surgery; Japan Nosocomial Infections Surveillance (JANIS) reported a 10% iSSI incidence in colorectal surgery in 2019 [2]. A high rate of iSSI in colorectal surgery has a negative impact on short-term outcomes, lengthens hospital stays, increases medical costs, and worsens esthetic outcomes [3,4]. Therefore, we conducted this study to ascertain whether preoperative oral antibiotics (OAs) can effectively reduce iSSIs. In the United States, in addition to oral antibacterial drugs, mechanical pre-treatment is also performed, but in Europe, mechanical pre-treatment is not currently performed [3,4].

## 2. Materials and Methods

### 2.1. Patients

This single institutional, retrospective, observational study comprised 620 patients with colorectal tumors who underwent surgery at our institution between January 2015 and December 2020. A total of 559 patients were included, after excluding patients with emergency surgery, multiple cancers, local recurrence, trans-anal tumor resection, infeasible preoperative mechanical bowel preparation (MBP), and OA use other than kanamycin sulfate or metronidazole. 

### 2.2. Ethical Approval and Consent to Participate

This study was conducted in compliance with the Ethical Guidelines for Medical Research Involving Human Subjects and was approved by the Ethics Committee of the Tochigi Cancer Center (approval number: 23-A053). Comprehensive informed consent for the use of clinical data was obtained from all eligible patients. Moreover, informed consent was also obtained from the parents and/or legal guardians of minor patients involved in this study (age under 18 years).

### 2.3. MBP and Preoperative OA

All patients were allowed to eat until noon the day before surgery and were given an oral rehydration solution the morning of surgery. For MBP, 24 mg of sennoside was administered 2 days before surgery, and 34 g of magnesium citrate was administered the day before surgery. In addition to MBP, OAs were started at our hospital in July 2017 (OA group). The OAs consisted of kanamycin sulfate 2 g/day and metronidazole 1 g/day administered orally in four divided doses from the day before surgery. 

### 2.4. Other Preventative Strategies for SSI and Surgical Procedures

Regarding the treatment of body hair, hair removal was performed using a clipper instead of shaving after anesthesia induction. One gram of cefmetazole was administered as a prophylactic antibiotic within 30 min before surgery, with additional doses administered every 3 h during surgery. Postoperative antibiotics were administered every 12 h and discontinued on the first postoperative day. The wound length was adjusted according to the tumor size. Protective wound-edge drapes were used during laparotomy, and anastomosis was performed using hand-sewn or mechanical anastomosis at the surgeon’s discretion. Doctors and nurses changed their gloves after the gastrointestinal anastomosis. We changed all surgical instruments to new ones before closing the wound. The skin and fascia were closed with absorbable monofilament sutures. Wound cleaning with high- pressure lavage was performed after fascial closure using 200 mL of saline for laparoscopic surgery and 500 mL in the case of open surgery. Antimicrobial-containing sutures were not used for abdominal wall sutures. Finally, dermally embedded sutures were used. These SSI prevention measures were performed on all eligible patients by all healthcare professionals, including doctors and nurses, who complied with standard precautions and hand hygiene based on the Centers for Disease Control and Prevention guidelines [5]. All surgeries were performed or supervised by surgeons who had sufficient experience and were certified by the Japan Society of Endoscopic Surgery and the Japanese Society of Gastrointestinal Surgery. Thirteen doctors were involved in surgery during the period in which this study was conducted. Colorectal resection with lymph node dissection was performed on eligible patients according to the 2019 Japanese Society for Cancer of the Colon and Rectum guidelines for colorectal cancer treatment [6]. All procedures were initiated at least 2 min after disinfection with iodine. 

### 2.5. Definition of SSI

iSSI was defined as an infection occurring within 30 days after surgery. A diagnosis of iSSI was made by the attending or primary surgeon according to the Japan Nosocomial Infections Surveillance (JANIS) criteria for SSI [7], with the addition of an infection control nurse. The main criteria of JANIS comprise infection of the incision occurring within 30 days after surgery and at least one of the following: (1) purulent drainage from the wound; (2) pathogen isolation from the incision; (3) wounds showing signs and symptoms of infection (but do not meet this criterion if the culture is negative); and (4) SSI diagnosed by the attending physician. SSIs were classified as iSSIs or organ/space SSIs (o/sSSIs). According to the CDC SSI guidelines, an iSSI is an SSI involving superficial incisions (skin and subcutaneous tissue of the incision) and deep incisions (fascia and muscles); an o/sSSI is classified as an anastomotic insufficiency or intra-abdominal abscess [8]. For incisional wounds, swab samples were collected from the draining or purulent drainage fluid. For organ cavities, puncture or drainage fluid was collected and cultured. The bacteria detected included Bacteroides fragilis, Eschrichia coli, and Enterobacter cloacae, and many of them were of intestinal origin. 

### 2.6. Statistical Analysis

A univariate analysis was performed using variables that categorized patients who underwent colorectal resection according to OA therapy (OA and non-OA). Student’s t-test was used for normally distributed continuous variables, and the Mann–Whitney test was used for non-normally distributed continuous variables. For categorical variables, the chi-squared test or Fisher’s exact test was used. A multivariate logistic regression analysis was performed to investigate the relationship between preoperative OA and SSI rates. Since this was a retrospective study, we used propensity score matching (PSM). PSM was used to perform a 1:1 match on the following nine factors in the two groups: sex, age, body mass index (BMI), American Society of Anesthesiologists performance status (ASA-PS), tumor location, presence of obstruction by the tumor, presence of smoking, presence of diabetes mellitus, and preoperative therapy. The match tolerance for the PSM was set to 0.20, and replacement was not allowed. All statistical tests were two-sided. *p*-values of <0.05 were considered significant. All statistical analyses were performed using the software package JMP Pro version 14 (SAS Institute Inc., Cary, NC, USA).

## 3. Results

The present study included 559 patients who underwent colorectal cancer surgery according to the inclusion and exclusion criteria. The patient characteristics are summarized in Table 1. The median patient age was 70 (23–95) years. The median BMI was 22.7 (13.5–41.09) kg/m^2^. Laparoscopic surgery was performed in 517 patients (92.5%), and open surgery in 42 patients (7.5%). Difficulty passing the endoscope owing to the tumor was observed in 73 of 559 cases (13.1). Thirty-three patients received chemoradiotherapy (*n* = 26), chemotherapy (*n* = 6), or radiotherapy (*n* = 1) as preoperative therapy.

There was one case of methicillin-resistant *Staphylococcus aureus* (MRSA) enteritis (0.3%), possibly due to OAs; however, there were no other cases of adverse reactions due to OAs. iSSI occurred in 37 patients (6.6%), and o/sSSI in 37 patients (6.5%), including 19 patients with anastomotic leakage (3.3%) (Table 2). 

The univariate and multivariate analyses of the relationship between iSSI and clinical factors are presented in Table 3. The multivariate analysis identified non-OA as an independent risk factor of iSSI (odds ratio [OR] 2.44, 95% confidence interval [CI]:1.22–4.88, *p* = 0.0112).

Of the 559 patients, 338 were included in the OA group. Before PSM, significant differences were observed between the OA and non-OA groups regarding difficulty passing the endoscope due to the tumor and ASA-PS (Table 4).

To eliminate these biases, PSM was used (Table 4). In the matched cohort, the iSSI rate was significantly lower in the OA group than in the non-OA group (iSSI rate, 1.61% vs. 5.07%; OR 3.4; 95% CI 0.123–0.707; *p* = 0.0062) (Table 5). 

## 4. Discussion

This study showed that not using MBP and OAs preoperatively increased the risk of iSSI 3.4 times compared with combined MBP and OA use in colorectal surgery classified as clean-contaminated wounds (Class II), based on the surgical wound infection classification grade defined by the CDC. The incidence of iSSI in colorectal (CRC) surgery in this study was reduced to 1.6% by administering preoperative OAs, which was markedly lower than those previously reported 3–20% [9,10]. Furthermore, the incidence of iSSI was lower than reported in a randomized controlled clinical OA trial [11,12]. The decrease in iSSI incidence may be attributed to improved perioperative management, including anesthesia management, and further reduction of the open abdominal wound by intracorporeal anastomosis [13]. Based on the results of this study, antimicrobial-containing sutures may not be essential for abdominal wall suturing. Two previous reports [14,15] described the effectiveness of preoperative OAs using PSM; however, the reports were limited to laparoscopic surgery or rectal cancer. This, to our knowledge, was the first study in Japan to describe the efficacy of OAs using PSM without limiting the surgical technique or site. 

Preoperative OA administration for CRC surgery was initiated by Garlock et al. [16] in 1939, and in recent years, the CDC guidelines have recommended the divided administration of non-absorbable OAs on the day before surgery [8]. However, according to a survey in Japan, preoperative OA administration rates are low, at only 9.7% [14]. This is due to the negative literature on combined OAs, such as the problem of MRSA enteritis caused by the administration of OAs in the past and the fact that preoperative prophylactic administration of OAs is not covered by insurance [17,18]. The CDC guidelines recommend OAs only on the day before surgery, and this recommendation has been strictly adhered to in random controlled trials performed in Japan in recent years, with no significant difference in the incidence of enterocolitis between the treated and untreated groups [11,12]. In fact, due to strict adherence to the CDC guidelines, MRSA enteritis was not observed at our institution. Although many centers do not administer preoperative OAs, even though it is now covered by insurance, it is essential to report the benefits of preoperative OAs, as in our case. 

Many studies have been conducted on the risk factors for iSSI after colorectal surgery, with notable associations reported for BMI, operation time, blood loss, surgical field contamination, and surgical techniques [19,20]. Laparoscopic surgery is less invasive than open surgery, and there are reports that it has fewer iSSIs [21,22]. In this study, a tumor in the rectum tended to correlate with iSSI occurrence, although not significantly. The rectum had a high incidence of iSSI in a previous report [7]. Risk factors for the occurrence of SSI in rectal cancer surgery include the distance from the anal verge and the creation of a stoma. We compared blood loss, transfusion volume, and operative time in the groups OA absence and OA presence. We found no significant difference in blood loss or transfusion volume between the two groups. Operating time was significantly longer in the group OA presence. We concluded that the improved surgical outcome was unrelated to SSI.

A strength of this study is its single-center design because the criteria for surgical quality, perioperative management, and iSSI were the same, and there was no inter-center bias.

However, its single-center design is also a limitation. This study was also retrospective with a small sample size, which may have limited its statistical power and introduced a statistical bias. Second, PSM was used to eliminate the effects of selection bias, although some residual confounding factors may not have been considered due to the retrospective observational design of this study. The longitudinal nature of the present study may have introduced a statistical bias between the two groups owing to changes in surgical techniques over the period included; however, no major changes were noted other than OA administration. Third, the possibility that the surgeon may have influenced the incidence of iSSI cannot be ruled out, as there were several doctors involved in the CRC surgery during the period of this study.

## 5. Conclusions

Incisional SSIs in colorectal surgery at our hospital were well controlled by preoperative OAs in addition to perioperative management and infection control. Combined OAs were effective in reducing iSSI during colorectal surgery.

## Figures and Tables

**Table 1 medicina-60-01970-t001:** Patient characteristics.

		*n* = 559 (%)
Sex	Males	324 (60%)
	Females	235 (42%)
Age, years	Median (range)	70 (23–95)
BMI (kg/m^2^)	Median (range)	22.7 (13.5–41.1)
ASA-PS	1	80 (14.3%)
	2	444 (79.4%)
	3	35 (6.2%)
Approach	Laparoscopic surgery	517 (92.4%)
	Laparotomy	42 (7.5%)
Location	Right	180 (32.2%)
	Left	379 (67.8%)
Colon/Rectum	Colon	308 (55.1%)
	Rectum	251 (44.9%)
Anastomosis	DST	295 (52.8%)
	Non-DST	264 (47.2%)
	Extracorporeal	204 (36.4%)
	Intracorporeal	60 (10.7%)
Obstruction	Yes	73 (13.1%)
	No	486 (86.9%)
History of smoking	Yes	334 (59.7%)
	No	225 (40.3%)
Diabetes	Yes	88 (15.7%)
	No	471 (84.3%)
Preoperative therapy	Yes	33 (5.9%)
	No	526 (94.1%)
Stoma	Yes	101 (18.1%)
	No	458 (81.9%)
OAs	Yes	338 (60.5%)
	No	221 (39.5%)

BMI, body mass index; ASA, American Society of Anesthesiologists; PS, physical status; DST, double stapling technique; OAs, oral antibiotics.

**Table 2 medicina-60-01970-t002:** SSI incidence outcomes.

iSSI	37 (6.6%)
o/sSSI	oSSI 10 (1.7%)
	sSSI 27 (4.8%)

iSSI, incisional surgical site infection, o/sSSI, organ/space surgical site infection.

**Table 3 medicina-60-01970-t003:** Univariate and multivariate analyses for detecting risk factors of incisional surgical site infection.

		Univariate Analysis					Multivariate Analysis		
		iSSI		OR	95% CI	*p*	OR	95% CI	*p*
		*n* = 37	%						
Sex	Males	25	4.47	1.5	0.76–3.15	0.2237			
Age, years	<70	19	3.4	1.06	0.54–2.07	0.8561			
BMI (kg/m^2^)	≥22.7	19	3.4	1.008	0.51–1.96	0.9811			
ASA-PS	ASA-PS1	7	1.25	1.43	0.60–3.38	0.4098			
Approach	Laparoscopic	6	1.07	2.61	1.02–6.67	0.0446	2.89	1.08–7.71	0.0733
Location	Left	30	5.37	2.12	0.91–4.93	0.0797			
Colon/Rectum	Rectum	26	4.65	3.12	1.50–6.44	0.0432	3.12	1.38–7.04	0.056
Anastomosis	DST	23	4.11	1.5	0.76–2.99	0.2392			
Obstruction	Yes	5	0.89	1.04	0.3–2.76	0.9323			
History of smoking	Yes	25	4.47	1.43	0.70–2.92	0.3178			
Diabetes	No	32	5.72	1.21	0.45–3.19	0.7005			
Preoperative therapy	Yes	5	0.89	2.75	0.99–7.61	0.0506			
Stoma	Yes	12	2.15	2.33	1.13–4.82	0.0219	1.23	0.54–2.79	0.601
OAs	No	22	3.94	2.38	1.20–4.69	0.0124	2.44	1.22–4.88	0.0112

BMI, body mass index; ASA, American Society of Anesthesiologists; PS, physical status; DST, double stapling technique; OAs, oral antibiotics; OR, odds ratio; CI, confidence interval.

**Table 4 medicina-60-01970-t004:** OA association with clinical factors after propensity matching.

Variables		Entire Cohort					After Propensity Score Matching				
		OA, yes		OA, no		*p*	OA, Yes		OA, No		*p*
		*n* = 338	%	*n* = 221	%		217	%	217	%	
Sex	Males	186	33.27	138	24.69	0.0959	134	30.88	134	30.88	>0.99
	Females	152	27.19	83	14.85		83	19.12	83	19.12	
Age, years	<70	176	31.48	103	18.43	0.2261	100	23.04	99	22.81	>0.99
	≥70	162	28.98	119	21.11		117	26.96	118	27.19	
BMI	<22.7	164	29.34	109	19.5	0.863	102	23.5	108	24.88	0.6311
	≥22.7	174	31.13	112	20.04		115	26.5	109	25.12	
Location	Colon	183	32.74	125	22.36	0.6023	129	29.72	124	28.57	0.697
	Rectum	155	27.73	97	17.17		88	20.28	93	21.43	
Obstruction	Yes	54	9.66	19	3.4	0.0143	22	5.07	19	4.38	0.7431
	No	284	50.81	202	36.14		195	44.93	198	45.62	
ASA	ASA-PS1	62	11.09	18	3.22	0.0008	15	3.46	18	4.15	0.7177
	ASA-PS≥2	276	49.37	203	36.31		202	46.54	199	45.85	
History of smoking	Yes	198	35.42	136	24.33	0.537	126	29.03	132	30.41	0.625
	No	140	25.04	85	15.21		91	20.97	85	19.59	
Diabetes	Yes	55	9.84	33	5.9	0.7223	30	6.91	33	7.6	0.7854
	No	283	50.63	188	33.63		187	43.09	184	42.4	
Preoperative therapy	Yes	16	2.86	17	3.04	0.1981	10	2.3	14	3.23	0.5295
	No	322	57.6	204	36.49		207	47.7	203	46.77	

BMI, body mass index; ASA, American Society of Anesthesiologists; PS, physical status; OA, oral antibiotic.

**Table 5 medicina-60-01970-t005:** Comparison of iSSIs with and without OA after PSM.

	OA Absence	OA Presence	OR	95% CI	*p*
iSSI	22 (5.07%)	7 (1.61%)	3.4	0.123–0.707	0.0062
o/sSSI	13 (3.05%)	0 (0%)	Not estimable	0	0.987

iSSI, incisional surgical site infection; o/sSSI, organ/space surgical site infection; OA, oral antibiotic; PSM, propensity score matching; OR, odds ratio; CI, confidence interval.

## Data Availability

All data generated or analyzed during this study are included in this published article.

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
