# Peer review of "Effect of Preoperative Oral Antibiotics for Prevention of Incisional Surgical Site Infection After Colorectal Surgery: A Propensity Score Matching Study"

_medicina, 2024, doi:10.3390/medicina60121970_

Round 1
Reviewer 1 Report
Comments and Suggestions for Authors
-
Maybe the authors could offer more information on why
SSI are greater for tumors of the
rectum than for other segments
of the colon.
-
It would be good to see what were the bacterial
strains that generated SSI-were they related
to the skin or the colon/digestive tract.
-
In the introduction the authors
should offer more information
on other options for mechanical
bowel preparation. What are the
standards in other regions such
as Europe or USA. The article
is focused on Japan but the
journal has worldwide accessibility .
-
More citations should be included
in the study as the subject is of
great interest and has been frequently
addressed since 1940 as the authors mention.
What was the progress in the incidence
of SSI over time. A few lines in comparing
SSI in open vs laparoscopic would also be good.
-
The English of the article needs to
be reviewed by an native speaker.
-
The aim is well defined and the
conclusions reflect this.
-
The work fits in the journal scope
as it addresses a problem
-
that should be aimed at a specific category of patients.
-
The structure is simple and
straight forward.
-
The article can be used as
such and can be transferred to clinical practice.
-
More recent citations should be included.
-
The article can be published with minor revision.
-
Overall, this paper contributes valuable insights
into infection control for colorectal surgeries,
advocating for the addition of OAs to MBP
protocols. Given the benefits observed,
this study supports a potential shift in
perioperative practices to lower SSI
rates, though further research across
diverse settings would solidify these findings

Comments on the Quality of English LanguageCan be improved
Author Response
Reviewer 1 Comments
Reviewer: 1
Maybe the authors could offer more information on why SSI are greater for tumors of the rectum than for other segments of the colon.
Response: Thank you for your useful comments on our manuscript.
We added, “Risk factors for the occurrence of SSI in rectal cancer surgery include the distance from the anal verge and the creation of a stoma.”.
It would be good to see what were the bacterial strains that generated SSI-were they related to the skin or the colon/digestive tract.
Response: Thank you for your useful comments on our manuscript.
We added, “The bacteria detected included Bacteroides fragilis, Eschrichia coli, and Enterobacter cloacae, and many of them were of intestinal origin.”.
In the introduction the authors should offer more information on other options for mechanical bowel preparation. What are the standards in other regions such as Europe or USA. The article is focused on Japan but the journal has worldwide accessibility .
Response: Thank you for your useful comments on our manuscript.
We added, “In the United States, in addition to oral antibacterial drugs, mechanical pre-treatment is also performed, but in Europe, mechanical pre-treatment is not currently performed (3,4).”.
More citations should be included in the study as the subject is of great interest and has been frequently addressed since 1940 as the authors mention. What was the progress in the incidence of SSI over time. A few lines in comparing SSI in open vs laparoscopic would also be good.
Response: Thank you for your useful comments on our manuscript.
We added, “Laparoscopic surgery is less invasive than open surgery, and there are reports that it has fewer iSSIs (21,22).”. We have added references 21 and 22.
The English of the article needs to be reviewed by an native speaker.
Response: Thank you for your useful comments on our manuscript.
We have proofread the English. We can also provide a receipt if necessary.
More recent citations should be included.
Response: Thank you for your useful comments on our manuscript.
We have added references 21 and 22.
Reviewer 2 Report
Comments and Suggestions for Authors
It gives me great pleasure to read this article on the use of oral antibiotics and bowel preparation before surgery, especially as outdated paradigms are still applied even in modern facilities. While I believe this paper is worthy of publication, I would like to bring a few aspects of the manuscript to the authors' attention for improvement.
The reference list is limited, and several cited papers are outdated. For example, reference No. 8 is from 31 years ago, yet is cited as supporting recent practices: "and in recent years, the CDC guidelines have recommended the divided administration of non-absorbable OAs on the day before surgery(8)." Similarly, reference No. 19 dates back to 1991 and is used to support the paper's conclusions. I encourage the authors to expand, refine, and update the reference list to include more recent publications.
Minor English revisions are needed throughout the manuscript. I noticed primarily spelling errors rather than syntactic issues, which may reflect a degree of fatigue rather than a lack of knowledge. For example, "calcification" was used instead of "classification" in "based on the surgical wound infection calcification grade defined by the CDC."
Regarding the statistical analysis, I observed that one exclusion criterion was emergency surgery, yet 54 patients had obstruction, a condition typically qualifying for emergency surgery unless they were all stented and subsequently operated on a few days later. Additionally, while the first two tables present descriptive statistics, the third table compiles all the inferential data. "P" is not defined in the legend of table nr 3, nor is there any mention of the propensity score calculation for each patient.
In conclusion, this paper presents an original idea, addresses a relevant clinical question, and has the potential to enhance surgical practices worldwide. I recommend this article for publication following some essential revisions
Author Response
Reviewer 2 Comments
Reviewer: 2
It gives me great pleasure to read this article on the use of oral antibiotics and bowel preparation before surgery, especially as outdated paradigms are still applied even in modern facilities. While I believe this paper is worthy of publication, I would like to bring a few aspects of the manuscript to the authors' attention for improvement.
The reference list is limited, and several cited papers are outdated. For example, reference No. 8 is from 31 years ago, yet is cited as supporting recent practices: "and in recent years, the CDC guidelines have recommended the divided administration of non-absorbable OAs on the day before surgery(8)." Similarly, reference No. 19 dates back to 1991 and is used to support the paper's conclusions. I encourage the authors to expand, refine, and update the reference list to include more recent publications.
Response: Thank you for your useful comments on our manuscript.
We have changed the reference for No. 8 to one from 2017. And, we have also added references 21 and 22.
Minor English revisions are needed throughout the manuscript. I noticed primarily spelling errors rather than syntactic issues, which may reflect a degree of fatigue rather than a lack of knowledge. For example, "calcification" was used instead of "classification" in "based on the surgical wound infection calcification grade defined by the CDC."
Response: Thank you for your useful comments on our manuscript.
We changed "calcification" to "classification".
Regarding the statistical analysis, I observed that one exclusion criterion was emergency surgery, yet 54 patients had obstruction, a condition typically qualifying for emergency surgery unless they were all stented and subsequently operated on a few days later. Additionally, while the first two tables present descriptive statistics, the third table compiles all the inferential data. "P" is not defined in the legend of table nr 3, nor is there any mention of the propensity score calculation for each patient.
Response: Thank you for your useful comments on our manuscript.
Since we were able to perform elective surgery by placing a colorectal stent, we have decided not to consider these cases as emergency surgery cases in this study. Regarding the second question, we did not understand the intent of the question. We apologise for any inconvenience caused.
Round 2
Reviewer 2 Report
Comments and Suggestions for Authors
Thank you for adressing the comments raised previously. I am happy to endorse this paper for publication